# Pediatric Inflammatory Multisystem Syndrome (PIMS) Did Occur in Poland during Months with Low COVID-19 Prevalence, Preliminary Results of a Nationwide Register

**DOI:** 10.3390/jcm9113386

**Published:** 2020-10-22

**Authors:** Magdalena Okarska-Napierała, Kamila M. Ludwikowska, Leszek Szenborn, Natalia Dudek, Anna Mania, Piotr Buda, Janusz Książyk, Katarzyna Mazur-Malewska, Magdalena Figlerowicz, Maciej Szczukocki, Beata Kucińska, Bożena Werner, Lidia Stopyra, Agnieszka Czech, Elżbieta Berdej-Szczot, Aneta Gawlik, Paulina Opalińska, Artur Mazur, Danuta Januszkiewicz-Lewandowska, Cezary Niszczota, Teresa Jackowska, Jacek Wysocki, Ernest Kuchar

**Affiliations:** 1Department of Pediatrics with Clinical Decisions Unit, Medical University of Warsaw, Żwirki i Wigury 63A, 02-091 Warsaw, Poland; natalia.dudek@gmail.com (N.D.); ernest.kuchar@gmail.com (E.K.); 2Department of Pediatric Infectious Diseases, Wroclaw Medical University, Chałubińskiego 2-2a, 50-368 Wroclaw, Poland; leszek.szenborn@umed.wroc.pl (L.S.); a9nieszka.czech@gmail.com (A.C.); 3Department of Infectious Diseases and Child Neurology, Poznan University of Medical Science, Szpitalna 27/33, 60-572 Poznan, Poland; amania@ump.edu.pl (A.M.); katarzynamelewska@ump.edu.pl (K.M.-M.); mfiglerowicz@gmail.com (M.F.); 4Department of Pediatrics, Nutrition and Metabolic Diseases, The Children’s Memorial Health Institute, Al. Dzieci Polskich 20, 04-730 Warsaw, Poland; piotrbuda@gmail.com (P.B.); j.ksiazyk@ipczd.pl (J.K.); 5Collegium Medicum, Jan Kochanowski University, al. IX Wieków Kielc 19A, 25-317 Kielce, Poland; mszczukocki@gmail.com; 6Department of Pediatric Cardiology and General Pediatrics, Medical University of Warsaw, Żwirki i Wigury 63A, 02-091 Warsaw, Poland; beata.kucinska@wum.edu.pl (B.K.); bozena.werner@wum.edu.pl (B.W.); cezary.niszczota@gmail.com (C.N.); 7Department of Infectious Diseases and Paediatrics, S. Zeromski Hospital in Krakow, Osiedle Na Skarpie 66, 31-913 Krakow, Poland; lidiastopyra@gmail.com; 8Department of Paediatrics and Paediatric Endocrinology, Upper-Silesian Paediatric Health Center School of Medicine in Katowice, Medical University of Silesia, Medyków 16, 40-752 Katowice, Poland; elaberdej@poczta.onet.pl (E.B.-S.); endo_sk6@sum.edu.pl (A.G.); 9Department of Pediatrics, Pediatric Endocrinology and Diabetes, University of Rzeszow, Lwowska 60, 35-301 Rzeszow, Poland; paulina.opal@gmail.com (P.O.); drmazur@poczta.onet.pl (A.M.); 10Department of Pediatric Oncology, Hematology and Transplantation, Poznan University of Medical Sciences, Szpitalna 27/33, 60-572 Poznan, Poland; 1962dj@gmail.com; 11Department of Pediatrics, The Medical Centre of Postgraduate Education, Cegłowska 80, 01-809 Warsaw, Poland; tjackowska@cmkp.edu.pl; 12Department of Preventive Health, Poznan University of Medical Science, Smoluchowskiego 11, 60-179 Poznan, Poland; jwysocki@ump.edu.pl; 132nd Infectious Diseases Ward, Children’s Hospital in Poznan, Nowowiejskiego 56/58, 61-734 Poznan, Poland; 14Department of Pediatric Cardiology, Regional Specialist Hospital in Wroclaw, Research and Development Center, Kamieńskiego 73a, 51-124 Wroclaw, Poland; pawel.tracewski@gmail.com; 15Department of Pediatric Infectious Diseases, Medical University of Bialystok, Waszyngtona 17, 15-274 Bialystok, Poland; 16Department of Pediatrics, Faculty of Medical Sciences in Zabrze Medical University of Silesia, 3-go Maja 13-15, 41-800 Zabrze, Poland; 17Department of Pediatrics, Institute of Medical Sciences, University of Opole, 45-040 Opole, Poland; 18Department of Pediatrics, University Clinical Hospital in Opole, 45-040 Opole, Poland; 19Department of Pediatrics, Gastroenterology and Nutrition, Faculty of Medicine Collegium Medicum, University of Warmia and Mazury in Olsztyn, Żołnierska 18a, 10-561 Olsztyn, Poland; 20Department of Pediatrics and Rheumatology, Specialist Hospital Antoniego Falkiewicza, Warszawska 2, 52-114 Wroclaw, Poland; 21Department of Pediatrics, Allergology and Cardiology, Provincial Polyclinical Hospital in Torun, Konstytucji 3 Maja 42, 87-100 Torun, Poland; 221st Department of Children’s Diseases, Subdepartement of Pulmonology and Allergology, Świętokrzyskie Center of Paediatrics, Grunwaldzka 45, 25-736 Kielce, Poland

**Keywords:** PIMS, MIS-C, SARS-CoV-2, COVID-19, Kawasaki disease, survey

## Abstract

Pediatric inflammatory multisystem syndrome (PIMS) is a new entity in children, likely associated with previous coronavirus disease 19 (COVID-19) infection. Most of the reports about PIMS come from countries particularly hit by the COVID-19 pandemic. Our aim was to investigate the nature of inflammatory syndromes in Poland (country with low COVID-19 prevalence) and to perceive the emergence of PIMS in our country. On 25 May 2020, we launched a nationwide survey of inflammatory syndromes in children for retrospective (since 4 March 2020) and prospective data collection. Up to 28 July, 39 reported children met the inclusion criteria. We stratified them according to age (<5 and ≥ 5 years old) and COVID-19 status. The majority of children had clinical and laboratory features of Kawasaki disease, probably non-associated with COVID-19. However, children ≥5 years of age had PIMS characteristics, and nine children had COVID-19 confirmation. This is, to our knowledge, the first report of the PIMS register from a country with a low COVID-19 prevalence, and it proves that PIMS may emerge in any area involved in the COVID-19 pandemic. In a context of limited COVID-19 testing availability, other risk factors of PIMS, e.g., older age, should be considered in the differential diagnosis of inflammatory syndromes in children.

## 1. Introduction

Since late April 2020, a growing set of articles has been published describing pediatric inflammatory multisystem syndrome (PIMS)—a new inflammatory entity in children, temporally and geographically associated with the coronavirus disease 2019 (COVID-19) pandemic [1,2,3,4,5,6,7,8,9]. The first definition of PIMS had been announced by the Royal College of Paediatrics and Child Health (RCPCH) on 1 May [10]. Multisystem inflammatory syndrome in children (MIS-C) is an alternative name proposed in the United States of America (USA) [11] and adopted by the World Health Organization (WHO) [12]. Unlike PIMS, the MIS-C definition requires confirmed SARS-CoV-2 infection or COVID-19 exposure. Approximately 1000 cases of PIMS and MIS-C have been reported as of July 2020, with the vast majority of reports from countries particularly hit by the COVID-19 pandemic. Epidemiological studies revealed that an abrupt increase in PIMS incidence occurs about four–five weeks after the peak of local COVID-19 cases [5,7].

Unusual clusters of severely ill children observed in the United Kingdom (UK), France or the USA may not occur in countries where COVID-19 is not as prevalent. Moreover, there are concerns that the remarkably severe clinical course of PIMS emerging from reports published so far, with 50–80% of children requiring intensive care unit (ICU) admission, may represent an extreme point of the broader spectrum of the post-infectious inflammatory response to COVID-19 [3,13]. Thus, in countries with a lower COVID-19 prevalence, we may expect PIMS to be more scattered in distribution, which needs to be investigated.

Poland (population over 37.5 M citizens, highly homogeneous society with predominant Caucasian race) had a relatively low COVID-19 prevalence. As of 28th June, nearly 44,000 confirmed COVID-19 cases had been registered [14]. The incidence rate in Poland was approximately 1000/1 M, almost 3-fold less than in France, over 4-fold less than in the UK or Italy and 12-fold less than in the USA [15] (Figure 1).The lower infectious rate might partially result from a lower testing rate (Poland, 52,101/1 M citizens). Thus, the discrepancy in testing does not correspond to the incidence rate (testing rate was similar in France, only 2-fold higher in Italy, 3-fold higher in the USA and 4-fold higher in the UK). In Poland, children accounted for 0.8–2.8% of all laboratory-confirmed cases, similar to other countries [16].

Our aims were: to investigate the nature of inflammatory syndromes in Poland during the COVID-19 epidemic and perceive the emergence of PIMS in our country. On 25 May, we launched nationwide surveillance of pediatric inflammatory syndromes: the MultiOrgan Inflammatory Syndromes COVID Related Study (MOIS-CoR). In this report, we present clinical and laboratory characteristics of the first 39 children with inflammatory conditions, including 9 confirmed PIMS, diagnosed over a period from 4 March (when the first case of COVID-19 in Poland was diagnosed) to 28 July 2020.

## 2. Experimental Section

The voluntary surveillance for retrospective (since 4 March) and prospective data collection was initiated under the National Consultant of Pediatrics auspices. Anonymized patient data from 34 pediatric hospitals from all over the country (Figure 2) were extracted from electronic and paper records and collected through an online form developed for that purpose. Before the surveillance was launched, reporting clinicians underwent online training, which included the current state of knowledge about PIMS and the unified diagnostic approach to such patients recommended by the study’s expert committee. Patient management was at the discretion of the relevant treating clinicians.

Ethical approval was obtained from the Bioethics Committee at Wroclaw Medical University (CWN UMW BW: 39/2020).

Inclusion criteria were:Patients who required hospitalization since 4 March. The end of the study will be defined by the declaration of the end of the COVID-19 pandemic by the World Health Organization; 0–18 years old;Diagnosed Kawasaki disease (KD) or incomplete (atypical) Kawasaki disease (aKD) or toxic shock syndrome (TSS) or macrophage activation syndrome (MAS) or unspecified inflammatory syndrome;Exclusion of other infectious and non-infectious causes that could be responsible for the disease;SARS-CoV-2 polymerase chain reaction (PCR) or serology result could have been positive or negative. Due to the limited availability and reliability of serologic testing, a proven or likely COVID-19 criterion was not a condition determining inclusion to the registry.

KD and aKD were defined following the American Heart Association (AHA) guidelines [17]. TSS was established based on modified criteria by the Centers for Disease Control and Prevention (CDC) [18,19]. MAS was diagnosed based on the Paediatric Rheumatology International Trials Organization (PRINTO) criteria for MAS classification in systemic juvenile idiopathic arthritis [20]. The definition of the inflammatory syndrome was based on the WHO MIS-C definition with the exclusion of SARS-CoV-2 confirmation [12]. Detailed inclusion criteria and case definitions are presented in Table 1.

For patients who met the inclusion criteria, we collected demographic data, past medical history, data on COVID-19 exposure, clinical symptoms, physical examination findings, laboratory, imaging and cardiologic tests results, treatment and outcome. Three independent, experienced researchers verified the fulfilment of inclusion criteria and the diagnoses.

Most of the findings were interpreted as descriptive and exploratory. Results are presented as counts and percentages for categorical data and medians and interquartile ranges (IQRs) for continuous data, according to COVID-19 evidence and age groups (below or at least five years of age). Five years of age was choosen as it is an edge for increased KD incidence. Groups were compared with Mann–Whitney and ANOVA Kruskal–Wallis tests where appropriate. Categorical data were analyzed using chi-square, Fisher exact test and chi-square for high contingency tables where appropriate. Statistical analyses were done with the use of Excel 2016 and Statistica 12 (Stat Soft). Results with *p*-value < 0.05 were considered statistically significant.

## 3. Results

Thirty-nine from 41 reported patients fulfilled the study inclusion criteria (Table 1). Among them: 29 (74%) were male; all 39 were Caucasian. The median age was 3.1 years (IQR 1.4–6.6) and median body mass index (BMI) was 15.7 kg/m^2^ (IQR 14.6–17.4). Six patients (15%) had coexisting chronic diseases—two patients had chronic heart disease, one of them also had asplenia, two others had acquired immunodeficiencies and two were obese (Table 2). Overall, nine patients (23%) had evidence of SARS-CoV-2 infection or exposure in a household setting. In 18 (46%) cases, SARS-CoV-2 status was unknown, and in the remaining 12 (31%) cases, exposure history and SARS-CoV-2 tests (PCR and serology) were all negative. Overall, 34 patients (87%) had KD diagnosed: 20 (51%) classic KD and 14 (36%) aKD; five patients had inflammatory syndrome, and no cases of TSS were reported. Ten (26%) patients developed MAS. A median of 8 days (IQR 6–10.5) of fever of 39.0 °C (IQR 39.0–40.0) was reported, and children were admitted to the hospital after a median of 4 days (IQR 2.0–7.5) from the onset of symptoms (Table 3). Reported symptoms and signs included: dermatological in 37 (95%), mucocutaneous in 30 (77%), neurological in 30 (77%), gastrointestinal in 24 (62%), respiratory in 16 (41%) and musculoskeletal in 15 (39%) patients. Clinical presentation differed depending on age group (<5 years of age vs. ≥five years of age) and SARS-CoV-2 status (Table 2 and Appendix A). Among nine COVID-19 confirmed cases, one child was younger than 5 years, and he fulfilled inflammatory syndrome criteria and developed a severe complication—giant coronary arteries aneurysms (z-score 13). This case was published elsewhere [21]. The majority of patients (27; 69%) underwent chest imaging—chest X-ray (CXR) or computed tomography (CT). Of this group, 10 (26%) had no abnormalities. Lung infiltrates were found in 12 (44%), interstitial changes in 4 (15%) and pleural effusion in 4 (15%) cases. Thirty-seven (95%) children had an echocardiogram performed. Six (16%) patients had coronary arteries dilations or aneurysms, including one with giant aneurysms. Coronary artery abnormalities persisted until discharge from the hospital in four of them. Four (10%) patients had pericardial effusion, and one (3%) had decreased contractility of the left ventricle with values of shortening fraction (SF) of 27% and ejection fraction (EF) of 52%. Laboratory findings according to age group and SARS-CoV-2 status are presented in Table 4 and in Appendix A. Only one child required treatment in an ICU and mechanical ventilation, and four (10%) other children needed oxygen supply. Thirty-five (90%) children were treated with intravenous immunoglobulin (IVIG), 15 (39%) with steroids, 4 (10%) with cyclosporine A and 1 (3%) with etoposide. Thirty-three (85%) children received acetylsalicylic acid (ASA), two (5%) heparin and one (3%) warfarin. Thirty (77%) children were discharged home without any complications thus far. No deaths were reported.

## 4. Discussion

PIMS case reports have been reported from many different countries, both with high and low COVID-19 prevalences. Our study is the first, to our knowledge, nationwide register of pediatric inflammatory diseases from a country of low COVID-19 prevalence. The study inclusion criteria covered a few clinical syndromes overlapping with PIMS in order to capture as many cases of the new entity as possible. Nine laboratory-confirmed cases of PIMS in our cohort prove that PIMS may emerge in any pandemic area.

The vast majority of children registered in our survey fulfilled KD or aKD diagnostic criteria. Their clinical characteristics and laboratory results were typical for this well-known inflammatory disease of childhood [3,17]. A substantial proportion of children in our cohort probably had KD non-associated with SARS-CoV-2 infection.

Due to an unknown SARS-CoV-2 status in nearly half of the patients, we performed stratification by age (Table 2, Table 3 and Table 4). It is well established that the KD incidence rate is higher in children younger than 5 years of age [17]. First cases of PIMS described in the literature were referred to as KD [2], but soon after that it became clear that these are two different entities and that PIMS is more prevalent in school-aged children [3,5,6,7]. We found that children over five years of age presented with several distinct features consistent with PIMS from previous reports [3,5,6]. Older children more frequently had gastrointestinal symptoms (79% vs. 52%), with a predominance of abdominal pain, nausea and vomiting. Musculoskeletal symptoms were also more prevalent in the older age group. Lymphadenopathy was observed more commonly in younger children. Children over five years of age had a significantly lower lymphocyte count (mean value in the range of lymphopenia) and much higher ferritin values than the younger group. We found the distribution of symptoms in separate age groups to be similar to what has been described for PIMS by Dufort et al. and Feldstein et al. [5,6].

The features characteristic of PIMS were even more definite when comparing COVID-19-positive vs. -negative patients (Table 4). Clinical presentation of the nine patients who had confirmed SARS-CoV-2 infection or exposure history was consistent with findings described in current reports [3,5,6,22,23]. These patients were older, with more common gastrointestinal involvement and headaches. Furthermore, compared to SARS-CoV-2-negative patients, they developed a lower lymphocyte count, platelet count and hyponatremia, and higher CRP and ferritin level (all of them statistically significant).

The lack of serologic evaluation in nearly half of our patients reflects its limited availability. Moreover, serologic SARS-CoV-2 diagnostics have their limitations. The concentration of antibodies against SARS-CoV-2 wanes rapidly over time and some patients never develop a detectable amount of antibodies [24,25,26]. The sensitivity and specificity of serological assays depend on the producer and are diverse [27]. The positive predictive value depends on a disease prevalence in society [28]. The unified international definition of PIMS or MIS-C has not been established yet. We suggest that a laboratory-confirmed SARS-CoV-2 infection should not be universally obligatory.

The exact incidence and risk of developing PIMS are challenging to assess. The estimated incidence of confirmed PIMS in Poland as of July was approximately 0.1 per 100,000 children and adolescents, which is 20 times lower than that reported in New York State by Dufort et al. [5]. The small number of COVID-19-related PIMS cases in Poland is another argument supporting that PIMS is a post-infectious complication of COVID-19 in children. Simultaneously, only one child (3% of all cases and 11% of COVID-19-confirmed cases) developed shock and required intensive care treatment. Due to the small number of patients in our report, no general conclusions should be drawn so far, but in countries with a high COVID-19 prevalence, even first reports showed a high proportion of children with shock and requiring ICU. If a lower rate of life-threatening complications in Polish patients with PIMS persists, a specific homogeneous racial and genetic background would be the most convincing explanation. All children reported in our study were Caucasian and we do not expect high ethnical diversity as racial minorities in Poland are very small. In other reports, Black African and Hispanic populations were significantly overrepresented among PIMS cases [3,5,6]. Current data support that genetic predisposition is associated with a more severe COVID-19 course, and that ethnic groups might have an association with PIMS [4,22]. The clinical presentation of children with confirmed PIMS in our group supports the broader range of PIMS manifestations [29]. However, given the small number of cases, this should be interpreted with caution. Higher vigilance of PIMS among clinicians involved in our register at the early stage of the pandemic may explain a higher number of benign cases.

### Limitations

The lack of historical data about the incidence and clinical characteristics of pediatric inflammatory diseases in Poland makes it impossible to compare our findings to pre-pandemic data. A relatively small group of patients in our register necessitates further surveillance to obtain more data and perform reliable statistical analyses. The number of health centers involved in the study is limited, but new units are still recruiting.

## 5. Conclusions

The major aspect of our study is that despite the low COVID-19 prevalence and homogenous racial background, we have captured PIMS emergence in our country. The incidence rate of PIMS corresponds to COVID 19 prevalence. The distribution of PIMS may be more scattered. Rising awareness of this new pediatric condition among clinicians is essential for prompt diagnosis and the appropriate approach to such patients. In the context of limited SARS-CoV-2 testing availability, other risk factors of PIMS, e.g., older age, should be considered in the differential diagnosis of inflammatory syndromes in children.

## Figures and Tables

**Figure 1 jcm-09-03386-f001:**
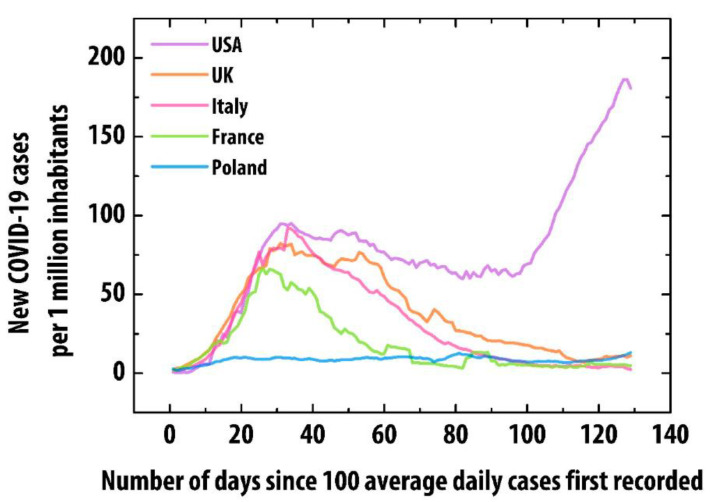
New COVID-19 cases per 1 million inhabitants in Poland compared with France, Italy, the United Kingdom (UK) and the United States of America (USA)—countries with reported cases of pediatric inflammatory multisystem syndrome (PIMS) [15].

**Figure 2 jcm-09-03386-f002:**
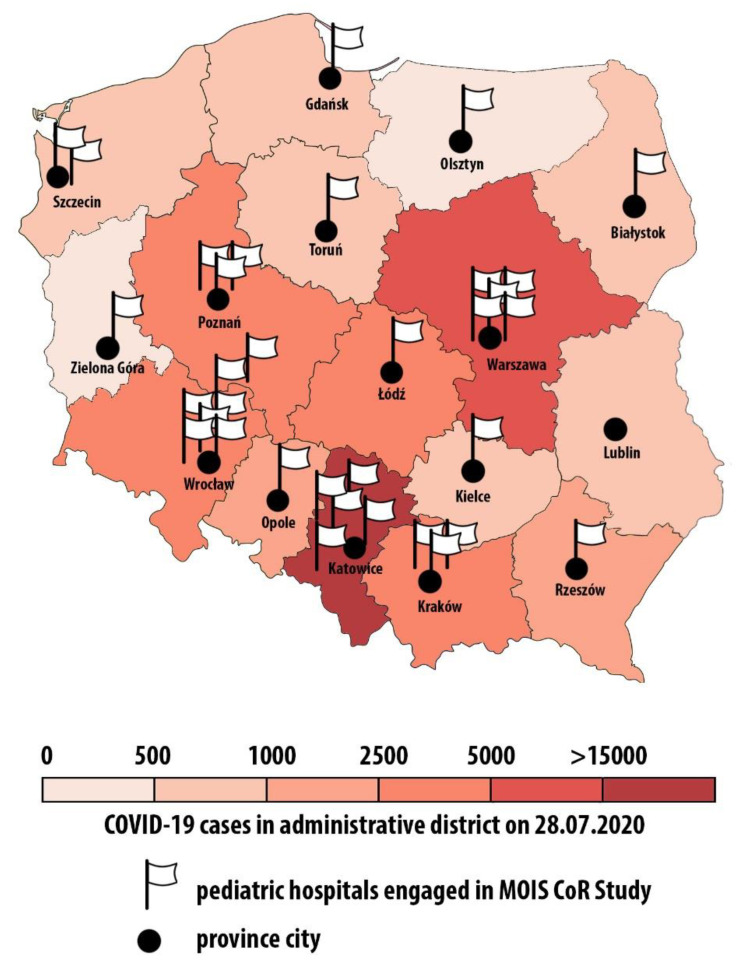
Pediatric hospitals engaged in the MultiOrgan Inflammatory Syndromes COVID Related Study (MOIS CoR Study)—Polish register of pediatric inflammatory syndromes during COVID-19 pandemic.

**Table 1 jcm-09-03386-t001:** Polish register of pediatric inflammatory syndromes (MOIS CoR Study): inclusion criteria.

Study Inclusion Criteria: Age, Disease Severity, Timing, Diagnosis Criterion and Exclusion of Other Causes Must Be Fulfilled.
Age: 0–18 years
Disease severity: requiring hospitalization
Time frame: since 4 March 2020 (ongoing)
Diagnosis: Kawasaki disease (KD) OR incomplete (atypical) Kawasaki disease (aKD) OR toxic shock syndrome (TSS) OR macrophage activation syndrome (MAS) OR unspecified inflammatory syndrome
Kawasaki disease (KD) case definition
Fever for at least 5 days and 4 from the following symptoms: (a) Erythema and cracking of lips, strawberry tongue and/or erythema of oral and pharyngeal mucosa (b) Bilateral bulbar conjunctival injection without exudate (c) Rash: maculopapular, diffuse erythroderma or erythema multiforme-like (d) Erythema and edema of the hands and feet in acute phase and/or periungual desquamation in subacute phase (e) Cervical lymphadenopathy (≥1.5 cm diameter)
Incomplete (atypical) Kawasaki disease (aKD) case definition:
Fever for at least 5 days and 2 or 3 from the above symptoms OR infant with unexplained fever for at least 7 days AND CRP ≥ 3 mg/dl and/or ESR ≥ 40 mm/h AND (1) At least 3 of the following: (a) Anemia for age (b) PLT ≥ 450,000 × 10^9^/L after the 7th day of fever (c) Albumin ≤ 3 g/dL (d) Elevated ALT (e) WBC count of ≥ 15,000 × 10^9^/L (f) Urine ≥ 10 WBC/hpf OR (2) Changes in echocardiogram suggesting KD
Toxic shock syndrome (TSS) case definition:
(1) Fever AND (2) Hypotension AND (3) At least two of the following organ systems involvement: (a) Gastrointestinal (vomiting, diarrhea, abdominal pain); (b) Muscular (severe myalgia, elevated creatine phosphokinase level); (c) Renal (sterile pyuria, elevated creatinine or urea); (d) Hepatic (elevated liver enzymes and/or bilirubin level); (e) Hematologic (decrease in PLT < 100 × 10^9^/L); (f) Disseminated intravascular coagulation; (g) Acute onset of diffuse pulmonary infiltrates and hypoxemia; (h) Acute onset of generalized edema, or pleural or peritoneal effusions with hypoalbuminemia; (i) Central nervous system (alterations in consciousness in absence of fever and hypotension)
Macrophage activation syndrome (MAS) case definition:
Febrile patient with: (1) Ferritin > 684 ng/mL AND (2) Any two of the following: (a) PLT ≤ 181,000 × 10^9^/L (b) AST > 48 U/L (c) Triglycerides >156 mg/dL (d) Fibrinogen ≤3 60 mg/dL
Inflammatory syndrome case definition:
(1) Fever for at least 3 days AND (2) High inflammatory markers (neutrophil count, CRP, ESR, procalcitonin) AND (3) Features of at least one organ dysfunction AND (4) At least two of the following symptoms: (a) Rash or bilateral non-purulent conjunctivitis or mucocutaneous inflammation signs (oral, hands or feet). (b) Hypotension or shock. (c) Features of myocardial dysfunction, pericarditis, valvulitis or coronary abnormalities (including echocardiographic findings or elevated Troponin/NT-proBNP) (d) Evidence of coagulopathy (by PT, PTT, elevated d-Dimers) (e) Acute gastrointestinal problems (diarrhea, vomiting or abdominal pain)
Exclusion of other infectious and non-infectious causes that could be responsible for the disease
SARS-CoV-2 testing may be positive or negative.

CRP: C-reactive protein; ESR: erythrocyte sedimentation rate; PLT: platelets; ALT: alanine transaminase; WBC: white blood cells; hpf: high power field; AST: aspartate transaminase; NT-proBNP: N-terminal pro B natriuretic peptide; PT: prothrombin time; PTT: partial thromboplastin time; SARS-CoV-2: severe acute respiratory syndrome coronavirus 2.

**Table 2 jcm-09-03386-t002:** Demographic and clinical characteristics of the patients, according to age group and SARS-CoV-2 status.

Characteristics	Overall (*N* = 39)	<5 Years (*N* = 25)	≥5 Years (*N* = 14)	*p*-Value	SARS-CoV-2 History Positive (*N* = 9)	SARS-CoV-2 History Unknown (*N* = 18)	SARS-CoV-2 History and Results Negative (*N* = 12)	*p*-Value
	**n, n/N or Med (% or IQR)**	**n, n/N or Med (% or IQR)**	**n, n/N or Med (% or IQR)**		**n, n/N or Med (% or IQR)**	**n, n/N or Med (% or IQR)**	**n, n/N or Med (% or IQR)**	
**Age [years]**	3.1 (1.4–6.6)	1.8 (0.85–2.8)	9.15 (6.4–12.15)	**<0.01 ***	10.5 (7–13.4)	3.3 (1.9–5.4)	1.2 (0.5–1.6)	**<0.01 ***
**Male sex**	29 (74%)	21 (84%)	8 (57%)	0.42	4 (44%)	13 (72%)	12 (100%)	**0.047 ***
**Any comorbidities**	6/39 (15%)	3/25 (12%)	3/14 (21%)	0.37	2/9 (22%)	3/18 (17%)	1/12 (8%)	0.62
**Clinical features**								
**Any dermatologic**	**37 (94%)**	**20 (80%)**	**13 (93%)**	0.39	**7 (78%)**	**17 (94%)**	**11 (97%)**	0.39
Rash	34 (87%)	22 (88%)	12 (86%)	0.60	6 (67%)	17 (94%)	11 (97%)	0.11
Hands and feet erythema or swelling	23 (59%)	13 (52%)	10 (71%)	0.20	7 (78%)	11 (61%)	5 (42%)	0.24
**Any mucocutaneous**	**30 (77%)**	**23 (92%)**	**10 (71%)**	0.11	**5 (56%)**	**15 (83%)**	**10 (83%)**	0.22
Conjunctivitis	26 (67%)	17 (68%)	9 (64%)	0.54	4 (44%)	13 (72%)	9 (75%)	0.27
Mucosal changes	27 (69%)	18 (72%)	9 (64%)	0.43	4 (44%)	15 (83%)	8 (67%)	0.12
Lymphadenopathy	20 (51%)	15 (60%)	5 (36%)	0.13	3 (33%)	9 (50%)	8 (67%)	0.32
**Any musculoskeletal**	**15 (39%)**	**6 (24%)**	**9 (64%)**	**0.01 ***	**5 (56%)**	**6 (33%)**	**4 (33%)**	0.48
Arthritis (swollen joints)	8 (21%)	2 (8%)	6 (43%)	**0.01 ***	4 (44%)	3 (17%)	1 (8%)	0.11
Arthralgia (without swelling)	9 (23%)	2 (8%)	7 (50%)	**<0.01 ***	3 (33%)	5 (28%)	1 (8%)	0.33
Myalgia	7 (18%)	3 (12%)	4 (29%)	0.19	3 (33%)	1 (6%)	3 (25%)	0.15
**Any gastrointestinal**	**24 (62%)**	**13 (52%)**	**11 (79%)**	0.09	**7 (78%)**	**10 (56%)**	**7 (56%)**	0.52
Nausea or vomiting	15 (40%)	6 (24%)	9 (64%)	**0.02 ***	5 (56%)	6 (33%)	4 (33%)	0.48
Abdominal pain	16 (42%)	6 (24%)	10 (71%)	**<0.01 ***	5 (56%)	7 (39%)	4 (33%)	0.22
Diarrhea	13 (33%)	8 (32%)	5 (36%)	0.54	5 (56%)	4 (22%)	4 (33%)	0.22
**Any neurologic**	**30 (77%)**	**21 (84%)**	**10 (71%)**	0.29	**5 (56%)**	**13 (72%)**	**11 (92%)**	0.16
Neck stiffness	3 (8%)	2 (8%)	1 (7%)	0.71	1 (11%)	0	2 (17%)	0.61
Somnolence	17 (44%)	12 (48%)	5 (36%)	0.34	4 (44%)	6 (33%)	7 (58%)	0.39
Headache	6 (15%)	0	6 (43%)	**0.001 ***	5 (56%)	1 (6%)	0	**<0.01 ***
**Any respiratory**	**16 (41%)**	**14 (56%)**	**4 (29%)**	0.09	**4 (44%)**	**5 (28%)**	**7 (58%)**	0.24
Cough	6 (15%)	4 (16%)	2 (14%)	0.63	2 (22%)	2 (11%)	2 (17%)	0.74
Chest pain	1 (3%)	0	1 (7%)	-	1 (11%)	0	0	-
Dyspnea	4 (10%)	2 (8%)	2 (14%)	0.45	2 (22%)	0	2 (17%)	0.42

BCG: Bacillus Calmette–Guérin vaccine; BMI: body mass index; IQR: interquartile range, med: median. We have listed major clinical findings in Table 2. Detailed clinical characteristics are presented in Appendix A. * statistically significant (*p* < 0.05).

**Table 3 jcm-09-03386-t003:** Clinical course and outcomes, according to age group and SARS-CoV-2 status.

Characteristics	Overall (*N* = 39)	<5 Years (*N* = 25)	≥5 Years (*N* = 14)	*p*-Value	SARS-CoV-2 History Positive (*N* = 9)	SARS-CoV-2 History Unknown (*N* = 18)	SARS-CoV-2 History and Results Negative (*N* = 12)	*p*-Value
	**n, n/N or Med** **(% or IQR)**	**n, n/N or Med** **(% or IQR)**	**n, n/N or Med** **(% or IQR)**		**n, n/N or Med** **(% or IQR)**	**n, n/N or Med** **(% or IQR)**	**n, n/N or Med** **(% or IQR)**	
Time from symptom onset to hospital admission [days]	4.0 (2.0–7.5)	5.0 (2.0–9.0)	4.0 (2.5–6.0)	0.37	4.0 (2.0–6.0)	5.0 (2.25–9.0)	4.5 (2.75–5.25)	0.56
	SARS-CoV-2 epidemiological data							
Confirmed contact with COVID-19	6/29 (21%)	0/17	6/12 (50%)	**<0.01 ***	6/8 (75%)	0/12	0/9	-
SARS CoV2 RT-PCR test positive	1/34 (3%)	0/21	1/13 (8%)	-	1/9 (11%)	0/13	0/12	-
SARS CoV-2 serology test positive ^a^	6/21 (29%)	1/13 (8%)	5/8 (63%)	**0.011 ***	6/9 (67%)	0/0	0/12	-
	Diagnosis							
Classic KD	20 (51%)	13 (33%)	7 (50%)	0.58	4 (44%)	10 (56%)	6 (50%)	0.85
Incomplete (atypical) KD	14 (36%)	10 (26%)	4 (29%)	0.36	3 (33%)	6 (33%)	5 (42%)	0.88
MAS	10 (26%)	3 (12%)	7 (50%)	**0.014 ***	5 (56%)	4 (22%)	1 (8%)	**0.04 ***
Inflammatory syndrome	5 (13%)	2 (8%)	3 (21%)	0.24	2 (22%)	2 (11%)	(8%)	0.61
	Therapy							
Intensive care treatment	1 (3%)	0	1 (7%)	-	1 (11%)	0	0	-
High-flow nasal cannula	5 (13%)	3 (12%)	2 (14%)	0.60	3 (33%)	0	2 (17%)	0.17
IVIG	35 (90%)	24 (96%)	11 (79%)	0.12	7 (78%)	16 (89%)	12 (100%)	0.61
GCS	15 (39%)	5 (20%)	10 (71%)	**<0.01 ***	7 (78%)	6 (33%)	2 (17%)	**0.02 ***
GCS and IVIG	14 (36%)	5 (20%)	9 (64%)	**<0.01 ***	6 (67%)	6 (33%)	2 (17%)	0.06
Cyclosporine A	4 (10%)	1 (4%)	3 (21%)	0.12	2 (22%)	2 (11%)	0	0.61
Etoposide	1 (3%)	0	1 (7%)	-	0	1 (6%)	0	-
ASA	33 (85%)	24 (96%)	9 (64%)	**0.016 ***	6 (67%)	15 (83%)	12 (100%)	0.29
Heparin	2 (5%)	1 (4%)	1 (7%)	0.59	1 (11%)	1 (6%)	0	-
Warfarin	1 (3%)	1 (4%)	0	-	1 (11%)	0	0	-
	Clinical course and outcome							
Shock	1 (3%)	0	1 (7%)	-	1 (11%)	0	0	-
Coronary arteries dilations or aneurysms ^b^	6 (15%)	3 (12%)	3 (21%)	0.36	3 (33%)	2 (11%)	1 (8%)	0.23
Discharged without complications ^c^	30 (77%)	19 (76%)	11 (79%)	0.57	4 (44%)	15 (78%)	11 (92%)	**0.03 ***

ASA: acetylsalicylic acid; GCS: systemic glucocorticoids; IQR: interquartile range; IVIG: intravenous immunoglobulins; KD: Kawasaki disease; MAS: macrophage activation syndrome, med: median. ^a^ Serology test was considered positive if any result of IgG or IgM was positive; there were 5 IgG positive results (one of them had also IgA antibodies against SARS-CoV-2) and one IgM and IgA positive result. ^b^ Coronary arteries dilations or aneurysms were defined on the basis of the American Heart Association recommendations for KD [17]. ^c^ No deaths were reported; four children remain under cardiologic care due to persistent dilations or aneurysms of coronary arteries; one patient was unable to walk for 4 weeks due to lower extremities pain, but returned to normal functioning; one patient with neurological complications had right side paresis, 1 remains under control due to cholestatic hepatitis and 2 are still treated for MAS. * statistically significant (*p* < 0.05).

**Table 4 jcm-09-03386-t004:** Laboratory results, according to age group and SARS-CoV-2 status.

Characteristics ^a^		Overall (*N* = 39)	<5 Years (*N* = 25)	≥5 Years (*N* = 14)	*p*-Value	SARS-CoV-2 History Positive (*N* = 9)	SARS-CoV-2 History Unknown (*N* = 18)	SARS-CoV-2 History and Results Negative (*N* = 12)	*p*-Value
		Med (IQR)	Med (IQR)	Med (IQR)		Med (IQR)	Med (IQR)	Med (IQR)	
White-cell count [×10^9^/L]	Min	8.0 (5.4–10.6)	8.2 (6.5–11.0)	5.5 (4.1–8.6)	**<0.01 ***	6.5 (4.0–10.4)	7.0 (5.6–9.3)	9.2 (7.0–11.1)	0.77
	Max	18.0 (4.6–22.6)	17.8 (15.2–22.6)	18.1 (10.9–23.2)	0.96	18.0 (14.8–23.9)	17.9 (13.0–20.4)	18.4 (15.3–23.2)	0.89
Lymphocytes count [×10^9^/L]		2.0 (1.05–3.5)	3.1 (2.0–4.94)	0.9 (0.6–1.1)	**<0.01 ***	0.8 (0.6–1.1)	2.0 (1.1–2.8)	4.5 (2.8–5.7)	**<0.01 ***
Platelet count [×10^9^/L]	Min	236.0 (150.0–465.0)	306.0 (203.0–530.0)	175.5 (125.5–298.0)	0.07	164.0 (124.0–180.0)	296.0 (195.3–455.8)	359.5 (225.5–549.0)	0.15
	Max	637.0 (495.0–830.5)	693.0 (530.0–862.0)	558.0 (413.0–727.6)	0.16	488.0 (328.0–522.0)	668.5 (541.6–769.5)	768.5 (616.3–980.6)	**0.04 ***
CRP [mg/dL]		129.2 (76.7–177.9)	128.4 (100.9–161.5)	132.0 (74.3–202.0)	0.53	190.0 (131.0–246.9)	107.0 (72.5–134.5)	128.8 (89.9–175.8)	**0.03 ***
Ferritin [ng/mL]		352.6 (156.2–1867.8)	164.0 (90.7–565.0)	1335.0 (352.6–9230.5)	**0.02 ***	1314.0 (330.0–3097.0)	1458.0 (133.9–27908.0)	175.5 (119.0–355.8)	0.37
D-dimer [mg/L]		3.6 (2.3–7.8)	3.5 (1.6–4.9)	2.9 (2.3–23.4)	0.36	2.9 (2.5–48.3)	2.3 (1.4–5.6)	4.2 (3.4–5.2)	0.38
Albumin [g/dL]		3.1 (2.6–3.5)	3.1 (2.8–3.4)	3.3 (2.5–3.5)	0.87	2.9 (2.4–3.5)	3.3 (3.1–3.6)	3.0 (2.7–3.2)	0.09
ALT [U/L]		32.0 (16.0–96.0)	19.0 (15.0–55.0)	69.5 (27.5–214.8)	**0.04 ***	32.0 (16.0–178.0)	51.5 (19.0–100.5)	23.0 (14.5–57.7)	0.27
Sodium [mmol/L]		135.0 (132.0–136.0)	135.3 (132.0–136.0)	134.5 (129.5–138.3)	0.43	131.0 (129.0–134.0)	136.0 (132.3–139.0)	135.5 (133.0–136.0)	**0.01 ***

ALT: alanine transaminase; CRP: C-reactive protein; IQR: interquartile range; med: median. ^a^ For CRP, ferritin, D-dimer and ALT levels—highest results were obtained; for lymphocytes count, albumin and sodium levels—lowest results were obtained. * statistically significant (*p* < 0.05).

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
