# Peer review of "Pediatric Inflammatory Multisystem Syndrome (PIMS) Did Occur in Poland during Months with Low COVID-19 Prevalence, Preliminary Results of a Nationwide Register"

_jcm, 2020, doi:10.3390/jcm9113386_

Round 1

Reviewer 1 Report

While it would have been interesting to have more detailed evidence of SARS-CoV-2 infection by either RNA PCR testing or serology, or a more robust history of COVID-19 exposure, the authors clearly outlined the limitations in doing so in the study. I think this is very important data from a country with low numbers of COVID-19 infection, and low number of children with PIMS or MIS-C

Recommendations:

1. Table 3:

Confirmed contact with COVID-19: Overall 6/29, but under SARS CoV2 history positive it says 7/8. Please reconcile

2. For the discussion, you could clarify that the lab conformed cases of PIMS were the 9, in the beginning of the discussion.

Author Response

Thank you for giving us the opportunity to submit a revised draft of the manuscript “Pediatric inflammatory multisystem syndrome (PIMS) did occur in Poland during months with low COVID-19 prevalence. Preliminary results of a nationwide register” for publication in the Journal of Clinical Medicine. We appreciate the time and effort that you dedicated to providing feedback on our manuscript and are grateful for the insightful comments on and valuable improvements to our paper. We have incorporated your suggestions. Please see below for a point-by-point response to the comments and concerns.

1)          Reviewer: While it would have been interesting to have more detailed evidence of SARS-CoV-2 infection by either RNA PCR testing or serology or a more robust history of COVID-19 exposure, the authors clearly outlined the limitations in doing so in the study. I think this is very important data from a country with low numbers of COVID-19 infection, and a low number of children with PIMS or MIS-C

Authors’ Response: Thank you!

2)          Reviewer: Confirmed contact with COVID-19: Overall 6/29, but under SARS CoV2 history positive it says 7/8. Please reconcile

Authors’ Response: Thank you for pointing this out. Overall 6 children had contact with COVID-19, we have corrected that under “SARS-CoV-2 history positive” in Table 3.

3)          Reviewer: For the discussion, you could clarify that the lab conformed cases of PIMS were the 9, at the beginning of the discussion.

Authors’ Response: We have modified the beginning of the discussion accordingly.

Reviewer 2 Report

The authors represent a large Polish consortium aiming to better describe PIMS all over Poland districts. Over a 5-month period, they could include and describe 39 patients with PIMS related or unrelated to Covid-19. Poland has experienced a low rate of SARS-CoV-2 infections, and the authors aim to analyze PIMS cases. 

The introduction and the Methods of the study are well presented. However, the study suffers from too many unanalyzed data in the Results section of the manuscript and from a limited interest of the discussion section. In my opinion, they could largely improve the attractively of the manuscript by better presenting their results. 

  • The main limitation of the study (i.e. the absence of pre-Covid-19 register of PIMS cases) should be exposed as a preliminary  data in the introduction section unseated of at the end of the study
  • The age group choice should be explained: why 5 years-old and not 6 years-old as Dufort et al. (if there is a reason) ?
  • Tables 2, 3, 4: very very hard to read. Far too many data, some of low importance. And p-values are missing in Tables 2 and 3. 
  • Table 2: too long. Consider keeping only the bold clinical data (and others in Supplemental data)?
  • Table 3: The diagnosis are different from the inclusion criteria. Please homogenize. The Covid-19 status is provided for 19 patients. What about the other 10 patients? 
  • Discussion section:
    • The nine patients with Covid-19 diagnosis finally meet the >5years group results in most data. Thus, are they different from the other >5years-old patients? In other terms, do you think that Covid-19 related PIMS only "added" 9 more patients to the register or that these 9 patients differed from other PIMS in severity or presentation?
    • Finally, were you surprised by the number of PIMS collected in 5 months? Is the incidence comparable to other countries (non-regarding Covid-19 status)? And was this study more interesting for PIMS description in Poland than for Covid-19 related PIMS?
    • Will the MOIS CoR Study group pursue in the future, even for non-Covid-19 related PIMS? 
    • Only one patient needed USI management, this is less than other reports. Could you comment?
  • Minor comments:
    • Table 4: Med Lymphocytes = 0 for the >5 group?
    • Intro and M&M: "error! Reference source not found"
    • Voivodeship is an unknown term for many readers. replace by "administrative district" or other more accurate terminology?
    • Inclusion criteria: avoid bullet numbers
    • Discussion: not statistically significant results should not be presented as so. A "tend" is hard to assess with little number of cases.

Author Response

Thank you for giving us the opportunity to submit a revised draft of the manuscript “Pediatric inflammatory multisystem syndrome (PIMS) did occur in Poland during months with low COVID-19 prevalence. Preliminary results of a nationwide register” for publication in the Journal of Clinical Medicine. We appreciate the time and effort that you dedicated to providing feedback on our manuscript and are grateful for the insightful comments on and valuable improvements to our paper. We have incorporated most of your suggestions. Please see below for a point-by-point response to the comments and concerns.

1)          Reviewer: The main limitation of the study (i.e. the absence of pre-Covid-19 register of PIMS cases) should be exposed as a preliminary  data in the introduction section unseated of at the end of the study

Authors’ Response: We do agree that the lack of pre-pandemic data is a potential limitation of our study, however, our main aim was not to compare pediatric inflammatory diseases during the COVID-19 pandemic to pre-pandemic data. Thus we respectfully disagree that the lack of pre-pandemic data is of such importance that it should be exposed in the introduction.

2)          Reviewer: The age group choice should be explained: why 5 years-old and not 6 years-old as Dufort et al. (if there is a reason)?

Authors’ Response: Thank you for this suggestion. The age threshold between analyzed groups was dictated by Kawasaki disease epidemiology with > 80% cases being < 5 years old. By separating children ≥ 5 years of age we aimed at extracting a subgroup in which COVID-related PIMS was more likely, bearing in mind limited availability of COVID-19 testing in Poland. We have explained this in the experimental section and in the discussion section.

3)          Reviewer: Tables 2, 3, 4: very very hard to read. Far too many data, some of low importance. And p-values are missing in Tables 2 and 3. Table 2: too long. Consider keeping only the bold clinical data (and others in Supplemental data)?

Authors’ Response: Thank you for pointing this out. We have rearranged the tables, leaving major findings in the manuscript and presenting more detailed data in the supplement. We have added p-values if relevant.

4)          Reviewer: Table 3: The diagnosis are different from the inclusion criteria. Please homogenize.

Authors’ Response: We have uniformed diagnoses given in the table according to those involved in inclusion criteria. There were no patients with toxic shock syndrome diagnosis.

5)          Reviewer: The Covid-19 status is provided for 19 patients. What about the other 10 patients?

Authors’ Response: While we appreciate the reviewer’s feedback, we respectfully disagree. COVID-19 status is provided for all patients as either positive (9 cases), unknown (18 cases), or negative (12 cases). Positive COVID-19 status was reported for patients with positive laboratory tests or exposure to COVID-19 case in history or both. Negative COVID-19 status applied to patients with both negative exposure history and negative PCR and serology tests. Unknown COVID-19 status referred to patients in whom no proof of COVID-19 exposure was found, but available data were incomplete (e.g. lack of serologic assessment). All these data are provided in both the results section and in Table 3.

6)          Reviewer: The nine patients with Covid-19 diagnosis finally meet the >5years group results in most data. Thus, are they different from the other >5years-old patients? In other terms, do you think that Covid-19 related PIMS only "added" 9 more patients to the register or that these 9 patients differed from other PIMS in severity or presentation?

Authors’ Response: Thank you for pointing this out. It is worth clarifying, that all but one “SARS-CoV-2 history positive” cases were also ≥5 years old. This explains why those two subgroups resembled each other in terms of clinical and laboratory findings.

As we have underlined in the Discussion section, the substantial proportion of patients in our register have likely had Kawasaki disease unrelated to COVID-19. Our aim was to identify COVID-related PIMS cases among them. The simplest way is laboratory testing for SARS-CoV-2, but this is not only of limited reliability but also frequently unavailable. Thus we stratified patients according to age to find out, that the ≥5 years old subgroup closely met COVID-19 positive subgroup data.

The number of children ≥5 years of age in our register is yet to small to divide it into subgroups and compare them reliably. However, some abnormalities in “SARS-CoV-2 positive history” subgroup were more profound than “≥5 years old” subgroup (e.g. inflammatory markers, high troponin and BNP, hypoalbuminemia), which suggests that COVID-positive patients were not just 9 more cases in the register, but made a distinct subgroup with the more severe course.

7)          Reviewer: Finally, were you surprised by the number of PIMS collected in 5 months? Is the incidence comparable to other countries (non-regarding Covid-19 status)? And was this study more interesting for PIMS description in Poland than for Covid-19 related PIMS?

Authors’ Response: We had not expected any defined incidence of neither Kawasaki disease nor other inflammatory conditions in Poland due to a lack of pre-pandemic data. According to published definitions, we understand PIMS as an entity temporally related to COVID-19. Our register aimed at capturing real PIMS cases (COVID-related) among a broader group of children with inflammatory conditions. Published data about PIMS incidence in other countries is still limited. Moreover, given the voluntary character of our register, the estimated PIMS incidence must be interpreted with caution.

The major aspect of our study is that despite low COVID-19 prevalence and homogenous racial background, we have captured PIMS emergence in our country.

8)          Reviewer: Will the MOIS CoR Study group pursue in the future, even for non-Covid-19 related PIMS?

Authors’ Response: We are planning to continue our project as long as the pandemic is ongoing. We have added this information in the methods section.

9)          Reviewer: Only one patient needed USI management, this is less than other reports. Could you comment?

Authors’ Response: There are a few potential explanations for this observation. Firstly, it may be due to statistical bias, as the number of confirmed PIMS cases in our register was small (9) and small numbers give extreme results. Secondly, this can be a consequence of training for clinicians preceding the surveillance, which could had raised their awareness of the disease, so that they have captured milder PIMS cases. Thirdly, it is possible, that in homogenous Caucasian society clinical presentation of PIMS may differ from that observed in more heterogenic populations of UK, France, or USA, but it is too early to conclude it with certainty. We have included these arguments in the discussion section.

10)        Reviewer: Table 4: Med Lymphocytes = 0 for the >5 group?

Authors’ Response: Thank you for pointing this out. The median lymphocyte count in this group was 0.9 ×109/L. We have corrected it.

11)        Reviewer: Intro and M&M: "error! Reference source not found"

Authors’ Response: We have not found any missing references nor errors in the manuscript file we have been sent by the Editor. Thus we were not able to correct it.

12)        Reviewer: Voivodeship is an unknown term for many readers. replace by "administrative district" or other more accurate terminology?

Authors’ Response: This has been corrected accordingly.

13)        Reviewer: Inclusion criteria: avoid bullet numbers.

Authors’ Response: We have corrected it accordingly.

14)        Reviewer: Discussion: not statistically significant results should not be presented as so. A "tend" is hard to assess with little number of cases.

Authors’ Response: Thank you for this suggestion. We do agree and have removed that.

Reviewer 3 Report

1). The premise that lower COVID rates in Poland compared to other counties would lead to a different phenotype of MIS-C does not make sense based on first principles. The incidence of MIS-C should clearly be lower in regions with less COVID, but it is not clear why the authors would propose that the clinical phenotype would differ.

2). The epidemiologic link between PIMS/MIS-C and prior COVID 19 exposure has been strongly established. The authors should revise statements questioning this established connection

3). The authors include a wide diversity of inflammatory illnesses (KD, MAS, suspected MIS-C) in this analysis so it is very difficult to draw any meaningful conclusions because the patient populations is too heterogenous. Additionally, presumably due to small sample size, there is minimal statistical analysis presented

4). I am not sure that the statement of this being the first report of MIS-C in a low COVID rate country is accurate. MIS-C has now been reported on all continents and in many countries. The author's need to confirm that there are no prior case reports that would contradict this statement

5). With only 9 cases of MIS-C, the authors should be careful to draw any conclusions about whether their patients had a milder course compared to prior reports

Minor comments:

1). what is the end date of the inclusion for this study

2). Sentences should not be started with numbers (i.e. like 39 should be Thiry nine)

3). Ejection fraction of 59% is normal. This patient should not have been classified as LV systolic dysfunction

Author Response

Thank you for giving us the opportunity to submit a revised draft of the manuscript “Pediatric inflammatory multisystem syndrome (PIMS) did occur in Poland during months with low COVID-19 prevalence. Preliminary results of a nationwide register” for publication in the Journal of Clinical Medicine. We appreciate the time and effort that you dedicated to providing feedback on our manuscript and are grateful for the insightful comments on and valuable improvements to our paper. We have incorporated most of your suggestions. Please see below for a point-by-point response to the comments and concerns.

1)          Reviewer: The premise that lower COVID rates in Poland compared to other counties would lead to a different phenotype of MIS-C does not make sense based on first principles. The incidence of MIS-C should clearly be lower in regions with less COVID, but it is not clear why the authors would propose that the clinical phenotype would differ.

Authors’ Response: Thank you very much for pointing this out. The different phenotypes of MIS-C might be expected in Poland, but not because of low COVID-19 prevalence, yet due to homogenous Caucasian genetic background. We have corrected this confusing sentence in the manuscript.

2)          Reviewer: The epidemiologic link between PIMS/MIS-C and prior COVID 19 exposure has been strongly established. The authors should revise statements questioning this established connection.

Authors’ response: We do agree that the link between PIMS/MIS-C and COVID-19 should not be questioned and we have corrected that in the manuscript.

3)          Reviewer: The authors include a wide diversity of inflammatory illnesses (KD, MAS, suspected MIS-C) in this analysis so it is very difficult to draw any meaningful conclusions because the patient populations is too heterogenous. Additionally, presumably due to small sample size, there is minimal statistical analysis presented.

Authors’ Response: Thank you for this comment. The major conclusion from our preliminary data is that PIMS/MIS-C has emerged in our country, despite low COVID-19 prevalence and that the incidence rate of PIMS corresponds to COVID 19 prevalence. Our register of a broad range of inflammatory disorders in children was aimed at capturing this new pediatric phenomenon among them. We have chosen only those inflammatory conditions which overlap with PIMS/MIS-C. Moreover, due to the limited availability and reliability of serologic tests for COVID-19, we stratified children according to age, hypothesizing that age criterion could appear helpful in identifying PIMS/MIS-C cases in such circumstances. However, we do understand that the numbers are yet too small to draw reliable conclusions and we have underlined it in the limitations section. We believe that clinical and laboratory data of confirmed PIMS/MIS-C in Poland, particularly milder clinical course of the disease as for now, broadens the knowledge about this new entity in children. We hope that our register would also encourage clinicians from other countries with lower COVID-19 prevalence to carefully monitor PIMS/MIS-C emergence in their societies.

4)          Reviewer: I am not sure that the statement of this being the first report of MIS-C in a low COVID rate country is accurate. MIS-C has now been reported on all continents and in many countries. The author's need to confirm that there are no prior case reports that would contradict this statement.

Authors’ Response: We do not state that we report the first MIS-C cases in a low COVID-19 rate country, but instead we emphasize that we present the first to our knowledge nationwide surveillance aimed at capturing MIS-C in a country with low COVID-19 prevalence. We have clarified this in the discussion section.

5)          Reviewer: With only 9 cases of MIS-C, the authors should be careful to draw any conclusions about whether their patients had a milder course compared to prior reports.

Authors’ Response: We do agree that the number of confirmed MIS-C cases is too small to draw firm conclusions. We have revised the discussion section in order to make it better-grounded.

6)          Reviewer: what is the end date of the inclusion for this study

Authors’ Response: The end date of the reported data is July, 28 – this had been written in the introduction. The study is still ongoing. The end of the study will be defined by the declaration of the end of COVID-19 pandemic by the World Health Organization. We have involved this information in the methods section.

7)          Reviewer: Sentences should not be started with numbers (i.e. like 39 should be Thirty nine)

Authors’ Response: Thank you, we have corrected that.

Round 2

Reviewer 2 Report

The authors have answered to my concerns, except:

1/ Table 2: still too long and not publishable in the current form. Please move some of the data in the Supplemental.

2/ You answer the following sentence to my number 7 comment. “The major aspect of our study is that despite low COVID-19 prevalence and homogenous racial background, we have captured PIMS emergence in our country”. In my opinion this sentence should be added to the manuscript conclusion.

Author Response

Thank you very much for your valuable comments!

  • Reviewer: Table 2: still too long and not publishable in the current form. Please move some of the data in the Supplemental.

Authors' Response: Thank you for this comment. We have rearranged the table so that major clinical findings are left in the manuscript, whereas detailed clinical characteristics are included in the Supplementary Materials (S1). We have also added the information about that in the Results section (lines 183-184) and beneath the Table 2. The numbering of the tables included in the Supplementary Material has thus changed – the table, which used to be S1, is now S2 (the references in the manuscript have been changed accordingly – lines 195 and 283-285).

  • Reviewer: You answer the following sentence to my number 7 comment. “The major aspect of our study is that despite low COVID-19 prevalence and homogenous racial background, we have captured PIMS emergence in our country”. In my opinion this sentence should be added to the manuscript conclusion.

Authors' Response: We have involved this sentence in the conclusions section (lines 268-269).

Reviewer 3 Report

The author's have improved the manuscript in the revised version.

Author Response

Reviewer: The author's have improved the manuscript in the revised version. Are the conclusions supported by the results? - Can be improved.

Thank you very much for your comment. We have modified the conclusion section in order to underline the major aspect of our findings presented in the results.